# Influence of the Gut Microbiome on Feed Intake of Farm Animals

**DOI:** 10.3390/microorganisms10071305

**Published:** 2022-06-27

**Authors:** Anna Grete Wessels

**Affiliations:** Institute of Animal Nutrition, Department of Veterinary Medicine, Freie Universität Berlin, 14195 Berlin, Germany; anna.wessels@fu-berlin.de; Tel.: +49-(0)30-838-63726

**Keywords:** appetite, eating behavior, farm animals, feed intake, intestine, microbiome, neurotransmitters, physiology, taste reception

## Abstract

With the advancement of microbiome research, the requirement to consider the intestinal microbiome as the “last organ” of an animal emerged. Through the production of metabolites and/or the stimulation of the host’s hormone and neurotransmitter synthesis, the gut microbiota can potentially affect the host’s eating behavior both long and short-term. Based on current evidence, the major mediators appear to be short-chain fatty acids (SCFA), peptide hormones such as peptide YY (PYY) and glucagon-like peptide-1 (GLP-1), as well as the amino acid tryptophan with the associated neurotransmitter serotonin, dopamine and γ-Aminobutyrate (GABA). The influence appears to extend into central neuronal networks and the expression of taste receptors. An interconnection of metabolic processes with mechanisms of taste sensation suggests that the gut microbiota may even influence the sensations of their host. This review provides a summary of the current status of microbiome research in farm animals with respect to general appetite regulation and microbiota-related observations made on the influence on feed intake. This is briefly contrasted with the existing findings from research with rodent models in order to identify future research needs. Increasing our understanding of appetite regulation could improve the management of feed intake, feed frustration and anorexia related to unhealthy conditions in farm animals.

## 1. Introduction

The field of microbiome research has evolved rapidly over the past few years and now calls for a holistic view of microbe–host interaction based on the holobiont theory [1]. This review relies on the definitions of Berg et al. [1], according to which the microbiota is the totality of living microorganisms in a certain ecosystem. While the term microbiome encompasses the community of microorganisms and their genes, the entire spectrum of molecules they produce, including their structural elements and metabolites. The microbiota produces an incalculable variety of metabolites, of which the best known are short-chain fatty acids (SCFA), branched-chain fatty acids, amino acids (AA) and biogenic amines, among many others. Intestinal microbes are capable of initiating the synthesis of incretins, hormones and neurotransmitters in their host [2]. This allows the microbiome to act on the animal and contribute to its metabolism, thus influencing health, welfare, productivity and certainly feed intake.

Some research on the influence of the microbiome on host nutrient intake was conducted in the context of obesity, mainly in rodent models. Laboratory rodent models provide the greatest possible standardization with respect to environmental factors that potentially influence the microbiome compared to large animal models. Individual mechanisms for how the microbiome influences eating behavior were identified in pigs [3]. It was shown that pigs with different feeding efficiencies might vary in the microbial community of their gastrointestinal tract (GIT). In this context, differences in voluntary feed intake were identified to cause the differential feed efficiencies [4]. These exemplary observations from various studies on pigs demonstrate that in times of highly efficient animal production, the microbiome of farm animals should also be considered in their feeding. Therefore, the purpose of this review is to provide a summary of the current state of research on the influence of the microbiome on the feed intake of farm animals.

## 2. Intestinal Microbiota in Farm Animals

Over millions of years of evolution, animals have developed the ability to host complex and dynamic consortia of microbes during their life cycle [5]. Colonization of the mammalian gut can begin as early as embryogenesis [6] and progresses to the formation of a complex and dynamic microbial community after birth [7]. In contrast, in birds, the first provision of feed initiates the simultaneous colonization of the microbiota in the different segments of the GIT [8]. The composition of the GIT microbiota depends on species, breed, age, nutrition, environment, rearing forms, stocking density, stress and especially antibiotics [9]. For this reason, the microbial patterns of various livestock species shown exemplarily in Table 1 should be considered with caution. The available literature on farm animals can only represent what has been produced by domestication and artificial rearing. It was proven that domesticated animals could not be compared to animals in a completely natural environment without limitations. For instance, the major bacterial families in domesticated pigs are *Lactobacillus* and *Enterobacteriaceae*, while wild *Suidae* were shown to have a high abundance of *Bifidobacterium* [10]. In commercial chicken lines for meat production, 36% of GIT microbiota species were affected by host genotype and sex [11]. Among them, 15 species affected belonged to the *Lactobacillus* genus. Rearing young chicken in a sanitized environment without contact with older conspecifics following the industrial in–out procedure leads to profound different microbiota compared to chicks kept with their adult hen for 24 h [12]. Moreover, in ducks, genotype affects *Lachnospiracecae*, *Bacteroidaceae* and *Desulfovibrionaceae* in the ceca, while overfeeding affects other families such as *Clostridiaceae*, *Lactobacillaceae*, *Streptococcaceae* and *Enterococcaceae* [13].

In general, the phyla Bacteroidetes, Firmicutes and Actinobacteria account for more than 90% of all known microbiota species in animals GITs. The remaining proportion is composed of Fusobacteria, Proteobacteria, Verrucomicrobia and Cyanobacteria [20,21,22]. The ratio of Firmicutes-to-Bacteroidetes is basically considered an indicator of the composition of the gut microbiome. Therefore, in human obesity research, a decreased Firmicutes-to-Bacteroidetes ratio was directly related to weight loss, whereas an increase in the ratio is associated with increased capacity for energy harvest from food [23]. The phyla Bacteroidetes and Firmicutes provide beneficial metabolites to the host from saccharolysis. Firmicutes species degrade polysaccharides to produce mainly butyrate, whereby species of Bacteroidetes break down complex carbohydrates to synthesize mainly propionate [24]. Other than fiber fermenters, proteolytic bacteria produce potentially beneficial but additionally harmful metabolites such as ammonia or hydrogen sulfide [25]. According to their food base, the ratio of saccharolytic to proteolytic bacteria depends on the supply of animals with dietary carbohydrates, including fiber and protein [26]. Ruminants are well adapted to rations rich in crude fiber due to their forestomach system. The breakdown of complex carbohydrates occurs in the anterior part of the GIT with the help of the microbiota, making nutrients and bacterial metabolites readily available to the animal for absorption in the intestine. In monogastric animals, crude fibers are not broken down in the foregut. The main fermentation organ is the posterior part of the large intestine. In pigs, complex carbohydrates can be digested exclusively and to a limited extent in the cecum and colon by bacterial means [27]. In chickens, the digestive capacity for crude fiber is even less pronounced since bacterial fermentation occurs almost exclusively in the ceca [9]. In accordance with the dietary basis, ruminants harbor more saccharolytic phyla than proteobacteria in relative proportion compared to monogastric animals (Table 1).

## 3. The Control of Feed Intake

### 3.1. Central and Peripheral Regulation

The mechanisms of hunger, satiation and satiety are highly complex processes controlled by the interaction of humoral factors and the central nervous system, as described in detail by Camilleri [28]. Briefly, the hypothalamus receives information from various peripheral (e.g., gastric distension, circulating nutrients and metabolites, hormones) but also central functional systems. In addition, sensory factors (such as aroma and the appearance of a ration, emotions and social variables) affect eating behavior. A key role in detecting and integrating peripheral feedback signals about the nutritional and metabolic status plays the arcuate nucleus (ARC), an aggregation of neurons in the mediobasal hypothalamus. The ARC receives peripheral signals either directly after crossing the blood–brain barrier (BBB) or indirectly via the afferent vagus and sympathetic nerves. The ARC contains two functionally antagonistic populations of neurons. In the anorexigenic neuronal population, cocaine- and amphetamine-regulated transcript (CART) and pro-opiomelanocortin (POMC) are coexpressed. Both can reduce appetite by releasing various anorexic signals in the brain, most notably α-melanocyte-stimulating hormone (α-MSH). By coexpressing the potent orexigens neuropeptide Y (NPY) and agouti-related peptide (AgRP), the orexigenic neuronal population increases appetite while inhibiting the anorexigenic-acting POMC-expressing neurons [29].

Peripheral signaling occurs via hormones secreted from endocrine cells in the GIT, such as cholecystokinin (CCK), glucagon-like peptide-1 (GLP-1), peptide YY (PYY) and ghrelin [29]. During the pre-absorptive phase, PYY, GLP-1 and CCK are secreted upon sensing nutrients such as glucose and AA by different receptors along the GIT. These acute satiety-inducing signals contribute to controlling meal size via the homeostatic and hedonic systems in the brain. Antagonistically, the stomach-derived ghrelin is a powerful stimulator of appetite whose secretion is decreased in response to food intake [25]. Numerous commensal and pathogenic bacteria synthetize peptides that are strikingly similar to leptin, ghrelin, PYY and NPY [30] and potentially affect the central regulation of appetite by triggering the respective neurons. Furthermore, stimulation of the aforementioned peripheral signaling pathways by microbial metabolites is a method of intervention of intestinal bacteria on the host that has been explored in rudimentary form so far. From rodent models, it is known that SCFA stimulates the secretion of GLP-1 and PYY [31,32]. In pigs, ileal infusions of the SCFAs, i.e., acetate, propionate and butyrate, increased the secretion of PYY [33] or plasma CCK levels [34]. During the post-absorptive phase, important peripheral feedback signals are released from the pancreas, such as insulin, glucagon and pancreatic peptide. The anabolic hormone insulin suppresses appetite and food intake via several central mechanisms, including insulin receptor-dependent induction of CART and α-MSH and decreasing expression of the orexigens NPY and AgRP. Together with insulin, another peripheral feedback signal originating from white adipose tissues, leptin acts as a long-term feedback signal in the hypothalamus to reduce appetite and food intake [25,29]. In the gut, microbial-derived SCFA stimulates leptin secretion [35]. Moreover, oral administration of a mix of SCFA caused an increase in plasma concentration of leptin in pigs [36]. Leptin crosses the BBB via a transport mechanism linked to the leptin receptor. In the ARC, leptin inhibits hunger signals such as NPY and AgRP but also stimulates POMC, which leads to the formation of the satiety signal α-MSH. Synergistic effects between leptin and CCK in the control of food intake were described by Voigt and Fink [37].

While the POMC/CART and NPY/AgRP neurons are often considered “first order” in the pathways of hunger and satiety, the major long-term positioners located in the hypothalamus are the mechanistic target of rapamycin (mTOR) and the adenosine monophosphate-activated protein kinase (AMPK). Both are directly involved in the long-term metabolic control of food intake that displays nutritional status, energy expenditure and body composition [38]. In the fed state, insulin and available nutrients (including AA) activate mTOR, which stimulates protein synthesis and inhibits autophagy. In the long-term, leptin increases hypothalamic mTOR activity, and the inhibition of mTOR signaling blunts leptin’s anorectic effect [38]. In contrast to mTOR, AMPK activity is increased during nutrient deficiency and inhibited by leptin and nutrient signals. Thus, AMPK and mTOR have overlapping and reciprocal functions and interact with the respective orexigenic or anorexigenic neurons [38].

Signals from the gut microbiome can potentially reach the CNS by several mechanisms: (1) direct activation of the vagus nerve or transmission of neurotransmitters; (2) production or induction of metabolites that pass the intestinal barrier, enter the bloodstream and may pass the blood–brain barrier to interfere with neurological function; and (3) microbial-associated molecular patterns (MAMPs, e.g., LPS) and metabolites produced by the microbiota may signal the immune system.

### 3.2. Neurotransmission and Reward Mediation

Food consumption beyond the coverage of nutrient requirements is caused by the rewarding nature of food as a stimulus to eat. The reward circuitry is complex due to the interaction of seven signaling systems (i.e., the opioid, dopaminergic, cannabinoid, GABAergic, serotoninergic, noradrenergic and neurotensin systems) as reviewed by Stanley et al. [39]. Some of the modulators, such as opioids or dopamine, stimulate a preferential appetite for palatable substances such as sugar or fat [40]. In contrast, stimulation of the serotoninergic system in the ARC causes an anorectic response even in the presence of palatable food since serotonin inhibits the orexigenic peptides (NPY/AgRP) and stimulates the anorexigenic (POMC, α-MSH) in response to peripheral signals such as CCK, ghrelin and leptin reaching hypothalamus via vagal afferents [37]. Homeostatic and hedonic mechanisms regulating food or feed intake are not independent, and while non-homeostatic eating is frequently attributed to the neurotransmitter dopamine, serotonin is largely seen as a neurotransmitter within the homeostatic system. Although enterochromaffin cells of the GIT are responsible for over 95% of the body’s serotonin production, the gut microbiota is understood to affect the host GIT serotonergic system [2]. For instance, spore-forming *Clostridium perfringens* can upregulate the expression of colonic tryptophan hydroxylase 1, boosting serotonin biosynthesis from tryptophan in the gut [41]. Further, the major bacterial metabolites such as SCFA directly induce serotonin production from enterochromaffin cells [41] and stimulate the release not only into the lumen of the gut but also into the vasculature. Besides serotonin, gut bacteria produce other neurotransmitters. Many transient and persistent inhabitants of the gut, including *Escherichia coli*, *Bacillus cereus*, *Bacillus mycoides*, *Bacillus subtilis*, *Proteus vulgaris*, *Serratia marcescens* and *Staphylococcus aureus* have been shown to produce dopamine [30]. *B. subtilis* appears to secrete both dopamine and norepinephrine into their environment, where it interacts with mammalian cells. *Bifidobacterium* spp. [42], *Escherichia* spp. [43] and *Lactobacillus* spp. were demonstrated to synthesize γ-Aminobutyrate (GABA) [44]. In pigs, GABA is assumed to be an orexigenic neurotransmitter [45]. Microbe-derived neurotransmitters are thought to serve host-microbe crosstalk (Figure 1).

The rewarding nature of food further includes the hedonic aspect. Food hedonics emerge from the confluence of energetic, cognitive and sensory signals. More commonly, however, hedonic value is singularly ascribed to a food’s taste, a multimodal perception arising from the central integration of taste, retronasal olfaction and oral somatosensation [46]. Most non-primate mammals, including humans, share the five primary taste sensations defined as sweet, umami, salty, sour and bitter [47]. Sweet taste identifies carbohydrates as an indicator of energy supply, umami recognizes AA representing dietary protein, salty indicates proper dietary electrolyte balance and sour and bitter identify potentially noxious and/or poisonous chemicals [47]. Taste chemosensing cells are epithelial cells allocated in taste buds along the oral cavity and the GIT. Pigs and cows count approximately 20,000 taste buds and therefore have nearly four times as many taste buds as humans [47]. In chickens and ducks, less than 500 taste buds were detected, and the sweet taste receptor as known from other species is missing [48]. Bitter, sweet and umami taste receptors belong to the superfamily of guanine-coupled nucleotide-binding protein-coupled receptors (GPCRs), class C, which was divided into two families: T1R and T2R, of which the first plays a substantial role for umami and sweet taste and can be additionally triggered by serotonin and the second for bitter taste sensation [49]. Apart from classical taste receptors, other GPCRs class C receptors for umami compounds were identified in taste cells [29]. The main substance eliciting umami taste is L-glutamate, which is widely present in food and, as such, represents protein, peptides and AA. Taste preferences may be influenced by altered taste receptor expression. Previously, it is known that there is genetic variation in taste perception due to natural allelic variations and common polymorphisms of taste receptors [50]. In murine enteroendocrine cell lines, SCFA enhanced the expression and activity of the umami taste receptors TASR1 and TASR3, thus altering the sensitivity of gut cells to bioactive nutrients [51]. In rainbow trout, bacterial, viral and parasitic infections were shown to alter the expression of T1R receptor genes infection [52]. In human, a positive correlation between taste sensitivity and some bacterial phyla of the oral cavity were found and it was assumed that taste responsiveness is affected by oral bacteria lining the tongue [53]. The potential influence of the microbiome on the expression of taste receptors and taste sensations should be considered, especially in livestock production. Opportunities from this newly discovered research field would be especially interesting from an animal welfare perspective but also economic targets as feed efficiency in farm animals.

## 4. Species-Specific Considerations

### 4.1. Monogastric Farm Animals

#### 4.1.1. Poultry

In poultry, the gut microbiota has already been shown to have effects on neurological processes such as anxiety and memory, as well as on the serotonergic system [54,55,56]. Nevertheless, poultry is less considered in research on the effects of the gut microbiome on eating behavior. Due to the short transit time, the colon is considered less important for microbe–host interactions [57]. In chickens and birds in common, the ceca are the site of the greatest intestinal fermentation [58,59]. Moreover, the crop contains a considerable amount of of microorganisms. This thin-walled, enlarged portion of the digestive tract of poultry, which serves to store food prior to digestion, contains primarily Firmicutes and Proteobacteria (~78% and ~16%, respectively) [8] but also significant amounts of lactobacilli [60]. Since efficient fermentation is correlated with a higher yield of nutrients for the host [61], some data are available on the association of cecal and fecal microbiota and feed efficiency in chicken. Bacteria belonging to *Bacteroides* [62], *Clostridium* [63], *Ruminococcus* [63,64], *Faecalibacterium* [63] and *Lactobacillus* [64,65] have been positively associated with feed efficiency in chickens, while bacteria belonging to *Enterobacteriaceae* [57], but more recently *Lactobacillus* [63] were also described to affect feed efficiency negatively. As the two directions of *Lactobacillus’* impact indicate, the findings for microbial patterns associated with high feed efficiency are inconsistent. This may be due to the sanitary measures in modern commercial hatcheries having an unwanted side effect of causing highly specific bacterial colonization of chicken’s intestines [66]. Other sources of variation are the dietary composition, the chicken line used and the encountered environmental microbes [63,64,67,68]. Further, Siegerstetter et al. [57] reported an association between the *Lactobacillus* genus and two *Lactobacillus crispatus* taxa and a high feed intake exclusively for female chickens. In general, there is a large impact of breeding on microbial colonization with an emphasis on *Lactobacillus* species [11]. In chickens, quantitative trait loci for the presence of bacteria such as *Lactobacillus* and *L. crispatus* colocalize with those for feeding behavior [69]. This colocalization suggests an influence of these indigenous bacteria on eating behavior, but this influence still needs to be strengthened by experiments with standardized intestinal microbiota inoculation and/or fewer influencing factors from the environment. However, the dietary addition of *Lactobacillus* strains clearly increases feed intake in chickens [67,68]. A possible mode of action might be the ability of *Lactobacillus* strains to synthesize GABA [44] since the release of GABA has been indicated to mediate the orexigenic effects of the hypothalamic NPY/AgRP signaling pathway [70].

The comparatively small amount of taste receptors and the absence of the previously known receptors for sweet taste suggest a lesser influence of flavor on feed intake in poultry. Even though chickens are seemingly insensitive to sweetness, dietary stevioside supplementation promoted feed intake through alteration of intestinal microbiota composition and the regulation of neuroactive pathways [71]. The authors concluded that stevioside might regulate the eating behavior through functional mechanisms other than its high-potency sweetness, such as decreased serotonin synthesis, enhancement of hypothalamic dopamine receptors and NPY signaling.

These few findings suggest that the gut microbiota influences eating behavior in poultry and justify the call for further research in this previously understudied group of farm animals.

#### 4.1.2. Pigs

For pigs, there is also limited evidence on the mode of action of the microbiota–host interplay regarding eating behavior. As described in poultry, different feeding efficiencies have been associated with variations in the GIT microbial community. Herewith the different feed efficiencies were caused by differences in voluntary feed intake [4]. In pigs, the cecum [4,72] and the colon [26,27] are considered the main microbial fermentation sites. McCormack et al. [73] reported that gut microbes associated with a leaner but healthier host (e.g., *Christensenellaceae, Oscillibacter* and *Cellulosilyticum*) were enriched in pigs with a low residual feed intake, a marker for higher feed efficiency. In the study of Metzler-Zebeli et al. [4], a higher abundance of *Campylobacter* in cecal mucosa was associated with the pig group of low residual feed intake, whereas *Escherichia*, *Shigella*, *Ruminobacter* and *Veillonella* were associated with pigs assigned to the high feed intake group.

According to Fleming et al. [3], individual mechanisms for how the microbiome influences eating behavior in pigs were identified to be colonic SCFA production, as well as peripheral concentrations of butyrate and serotonin. In pigs, ileal infusions of SCFAs (i.e., acetate, butyrate and propionate) increased the secretion of PYY [33] or plasma CCK levels [34]. The peptide hormone CCK was shown to induce satiety in pigs, as demonstrated in a feed motivation test [74]. Another mechanism for termination of feed intake is the SCFA-induced increase in plasma leptin concentrations, as shown by Jiao et al. [36].

Stimulation of the serotoninergic system in the ARC was shown to cause an anorectic response even in the presence of palate delights [37]. Thus far, it has been assumed that serotonin content in the porcine brain depends strongly on the uptake of tryptophan across the BBB, with implications on feed intake [75,76]. In pigs fed low protein diets, gut microbes produced notable amounts (0.3–2.0 g/d) of leucine, valine and isoleucine (further summarized as BCAA); and phenylalanine and lysine [77], thus contributing by 10% to the coverage of the estimated requirement of in pigs first-limiting lysine. The microbial synthesis of BCAAs has a higher proportion relative to the other AAs [78]. The BCAA leucine, but also the non-essential AA glutamine, are the most abundant constituents of plant and animal proteins and are produced in remarkable amounts by gut bacteria [79]. Besides their role in protein synthesis, these AAs individually activate the mTOR-signaling pathway to promote protein synthesis [80]. Since other AAs can activate mTOR exclusively insulin-depended, especially leucine seems to have a derived evolutionary function as an anabolic signal [81]. Glutamine amplifies glucose-stimulated insulin secretion, and with insulin, it can activate the mTOR pathway [82]. The activation of the hypothalamic mTOR signaling pathway induces an anorectic response [38]. Further, circulating leucine mediates the uptake of all large neutral AAs across the BBB [76] and also cerebral serotonin synthesis [75,83]. A primary leucine excess can thus lead to a secondary deficiency of valine [84] or tryptophan [76]. Pigs are capable of sensing AA imbalances, which in turn leads to a reduction in feed intake [84,85]. In the classical behavioral test for AA deficiency, pigs detect and reject a diet lacking an essential AA within 20 min following the onset of feeding [86]. This effect is autonomous from olfactory, taste or other peripheral systems [87]. The anterior piriform cortex (APC) is the behaviorally relevant chemosensor for essential AA depletion, projecting to neural circuits that control feeding [88]. Imbalances of circulating AA cause a decrease in the concentrations of a limiting essential AA in the APC. Initiation of mRNA translation occurs when AAs are acylated (=charged) to transfer ribonucleic acid (tRNA) by their cognate aminoacyl-tRNA synthetases. In the presence of essential AA deficiency, the cognate tRNA remains deacylated (=uncharged). Thus, AA deficiency leads to an accumulation of uncharged tRNA in the APC. The following cellular adaptation to AA deficiency is characterized by a decrease in global protein synthesis complemented by increased transcription of genes related to AA synthesis [88,89]. If the recognized deficiency and the subsequent neuronal signals cannot be responded to with a compensatory feed selection, the remaining AAs are metabolized and thus lost for protein biosynthesis [88]. In conventional pig farming, this natural reaction cannot be implemented due to stringent feeding practices and can thus have a reducing effect on feed intake. Gietzen and Rogers [90] reported that degradation of protein in the brain begins within 2 h after deficiency detection of a single indispensable AA. Therefore, this mechanism exists to prevent negative effects on the brain caused by AA deficiency. mTOR is not involved in sensing AA deficiency [89]. However, contributing to the production of the two pointer AAs, leucine and glutamine, the gut microbiota can potentially affect the host’s AA homeostasis, resulting in adaptations in eating behavior.

To date, no experimental studies have been published that have investigated a relationship between changes in the gut microbiota and the expression of the sense of taste in pigs. One study suggests modulation of the olfactory receptor OR51E1 by the gut microbiome and factors that affect the complexity of the microbiota [91]. For taste receptors so far, it has been shown that maternal antibiotic administration leads to an upregulation of the TAS1R1 gene for the umami taste receptor TAS1TR in the stomach of suckling piglets [92]. However, there is no evidence that increased expression of TAS1R1 could affect piglet appetite. It is more likely that this change in gene expression serves the host for the recognition of compounds produced by microorganisms present in the passing gastric bolus of the mother’s milk. The detection of umami taste is generally associated with the presence of glutamic acid. Glutamic acid could be evolutionarily considered as a marker molecule to sense the degree of protein digestion in the stomach [93]. The presence of the T1R1 and T1R3 genes in non-taste tissues of pigs suggests that the taste receptors may be involved in the chemosensory function of organs participating in several digestive, metabolic and behavioral processes. For this reason, there is a fundamental need for research on taste receptors in pigs and, more specifically, on the influence of the microbiota on these chemosensors.

### 4.2. Ruminant Farm Animals

Plant biomass, which is indigestible for monogastric animals, can be converted into digestible food by ruminants since their main fermentation organ is located in the anterior part of the GIT [94]. For this reason, previous studies in ruminants focused on the rumen, as energy production and nutrient supply to the host are considered a function of microbial fermentation in it [95]. It should be noted, however, that the functional and metabolic performances of the individual segments of the GIT are different and, in terms of the holobiont theory, contribute collectively to the health and nutritional status of the animal. The differences between each segment are reflected in their respective bacterial populations, which together may have an impact on host nutrition and energy balance [95,96]. In a study by Myer et al. [95], the majority of the ruminal bacteria of beef cattle belonged to the Bacteroidetes genus *Prevotella*, with the majority of the lower GIT taxa belonging to the Firmicutes genera *Butyrivibrio* and *Ruminococcus*. Rumen microbiota also includes large proportions of protozoa, archaea and fungi, which are essential to the rumen and host function [97]. Previously, the main mechanism for regulating feed intake in cattle was thought to be mechanical via rumen filling [98]. However, even in ruminants, recent studies indicate that microorganisms in the rumen may also contribute to the regulation of feed intake. Accordingly, cows with higher residual feed intake were associated with a greater ruminal relative abundance of *Ruminococcus gauvreauii* spp., while cows with lower feed intake had a greater ruminal relative abundance of *Howardella* [96]. Whereby *Ruminococcus gauvreauii* spp. is considered to be a fibrolytic bacterium, thus contributing to a higher fermentation rate of dietary fiber in the rumen and allowing a faster turnover of ruminal digesta and feed consumption in ruminants. In contrast, *Howardella* is a ureolytic bacterium that could promote rumen urea recycling in cows with lower feed intake to compensate for lower protein supply, thus increasing the efficiency of nutrient utilization. A less diverse but higher specialized rumen microbiome was proposed to promote the energy acquisition of Holstein Friesian milking cows, thereby improving their feed efficiency [94]. Other studies also showed that higher feed efficiency in ruminants is associated with the lower richness of the microbiome gene content and microbial taxa [97,99,100]. The apparent specialization of the microbial community in each of these microbiome groups resulted in better energy and carbon supply to the animal while reducing methane excretion. From this, the authors concluded that the more efficient microbiomes were less complex but more specialized to meet the energy needs of the host. However, this hypothesis is not clearly resolved in the literature, as it is refuted by other studies [101,102,103] that found no differences in microbial diversity between animals with high or low residual feed intake. On the other hand, the idea of a more efficient microbiota was supported by the finding of higher concentrations of SCFAs in rumen fluid [94], which can provide more than 70% of the energy requirements of ruminants after absorption [104]. Although the absorption characteristics of the rumen differ from mucosal surface-lined regions of the GIT, the transport of many molecules from the rumen, including SCFAs produced by intraruminal microbial fermentation, is well documented [105,106]. Because SCFAs play an important role in bovine energy homeostasis, they can be expected to act as effectors in regulating feed intake as described with other species.

Specific microbial genera may play important roles in the fermentative and cellulolytic capacity of the rumen based on their putative functions, and with this, they may contribute to the observed association between microbial community and feed efficiency. For instance, *Leucobacter* spp. holds genes that encode glycoside hydrolases and carbohydrate-binding modules targeting the breakdown of starch and oligosaccharides [99,100]. The butyrate producer *Butyrivibrio* can also ferment a variety of sugars, affecting the energy pool of enterocytes, of which butyrate is known to be a primary metabolic fuel [101,102]. High abundances of *Butyrivibrio* in ruminal fluids of dairy cows were associated with high residual feed intake [103]. The association of *Butyrivibrio* in jejunal samples of steers was contradictory and showed high feed efficiency due to lower feed intake and high daily gains [95]. *Dialister* is associated with hyposalivation [107], which may alter the buffering capacity of the rumen and fluid turnover as well as low methane emission. This is further supported by their association with a less stable pH in the rumen and higher abundancy in low methane emitters [108].

Changes in feeding behavior induced by the microbiota–host interaction are suspected in ruminants suffering from acidosis, which occurs with diets based on concentrates and insufficient fiber supply. The technique of rumen liquor transplantation can counteract acidosis and modify eating behavior [109]. Although pain relief or anti-inflammation may partially explain the effects on feed intake, the veterinary practice of liquor transplantation suggests that the rumen microbiota influences appetite in pathological conditions such as acidosis. Accordingly, in states of subacute acidosis in cows, ruminal microbiota is modified while feed intake, as well as the duration of rumination, are reduced [110]. The probiotic yeast *Saccharomyces cerevisiae* has a protective effect against acidosis-associated physiological changes, such as lowering rumen pH and changes in SCFA [111,112], and it has also been shown to produce behavioral changes, such as reducing the intervals between meals and a tendency to ruminate longer [113]. However, in healthy cattle, one of the main findings of Monteiro et al. [96] was that the composition of the rumen microbiome was dependent on feed intake and not vice versa. Furthermore, there were no significant associations between feed efficiency phenotypes and microbial communities in the cecum and colon at the phylum level [95].

In ruminants, in particular, the current lack of studies on microbial endocrinology, the physiological significance of microbially produced neurochemicals [114], and the dependence of taste receptor expression imply a strong need for research on regulatory mechanisms and studies beyond feed efficiency.

### 4.3. Rodent Models

The vast majority of knowledge on the regulatory mechanisms of microbial influence on food/feed intake to date was obtained from studies in rodent models. Transferability to monogastric livestock should be given but still needs to be confirmed by concrete studies. The extent to which transferability could be extended to ruminants remains to be elucidated.

In order to highlight the gap in knowledge on farm animals, the current status of research on rodent models was briefly summarized here. First, a rat model revealed that the microbiome affects the permeability of the gut barrier and the BBB and therefore influences the interaction between gut microbiota and the host’s brain unless the vagus nerve is involved [115]. Gut inoculation of mice with *Campylobacter jejuni* resulted in direct activation in the vagal sensory ganglia and the primary sensory relay nucleus for the vagus nerve, the *nucleus tractus solitarii* (NTS) [116]. The complex NTS contains a number of (sub)nuclei to which primary visceroafferent fibers of the facial nerve, the glossopharyngeal nerve and the vagal nerve project, and also in the upper section special nuclei of the sense of taste. The spatial proximity of the signal arrival from the intestine and the sense of taste allows the assumption that the interconnection of both signaling pathways could take place in order to establish corresponding behaviors, e.g., to learn taste preferences. Therefore, the NTS seems to be well adapted to the coordination of interoceptive feedback signals transmitted by the vagus nerve from the gut to the brain and from the brain to the periphery, thus acting as an excellent hub for microbiota–gut–brain signaling [2].

The main influences of the microbiota on the rodent host are based on the effects of SCFA [117]. The SCFA can increase the contractility of the colon in rats, as SCFAs exert their effects on the entire gut, enhancing nutrient absorption by their action on blood flow and accelerating transit through the colon [118]. Moreover, SCFAs can stimulate the secretion of GLP-1 and PYY [31,32], and microbial-derived indole was also shown to induce GLP-1 secretion [119]. Orexigenic ghrelin may be another link between the gut microbiome, gastrointestinal motility and appetite control, as variations in the gastrointestinal microbiome have been shown to influence ghrelin expression [120]. Another mechanism to affect host eating behavior is used by commensal *E. coli* via the caseinolytic protease B (ClpB) heat shock protein, an antigen mimetic of α-MSH. Chronic intragastric delivery of ClpB-expressing *E. coli* in mice stimulated the production of α-MSH-reactive antibodies and decreased feed intake [121]. Additionally, peripheral α-MSH was shown to trigger the release of PYY and GLP-1 from enteroendocrine L cells in the gut via activation of the melanocortin 4 receptor [122]. This suggests that PYY and GLP-1 could mediate the effects of bacterial ClpB on satiety.

Certain bacterial species such as *Clostridium perfringens* have pro-serotonin activity. For instance, serotonin concentrations were significantly reduced in the cecum and colonic lumen of germ-free mice [123]. Corresponding observations were also made for serotonin concentrations in the blood [78] and hippocampus of germ-free rats [124]. Accordingly, circulating levels of the precursor AA tryptophan appear to be dependent on microbial colonization. Germ-free mice showed variations in blood levels of tryptophan, which was compensated for after recolonization with cecal inocula from donor mice [125]. In another study using germ-free mice colonized with gut microbiota from three wild species with different foraging strategies (carnivore/insectivore, omnivore and herbivore mice), microbial colonization influenced the availability of tryptophan as well as isoleucine, phenylalanine and tyrosine [126]. In the same study, plasma tryptophan availability was significantly correlated with voluntary carbohydrate intake. In addition, germ-free rats had decreased hypothalamic histamine concentrations [127]. Moreover, a comparison of the cerebral metabolome of germ-free versus non-germ-free mice showed that 38 of the 196 metabolites analyzed were significantly altered [128]. As such, the administration of specific bacteria can result in altered metabolites of the central nervous system. Thus, *Bifidobacterium infantis* was shown to be able to decrease the concentrations of 5-hydroxyindoleacetic acid (5-HIAA) in the frontal cortex and of 3,4-dihydroxyphenylacetic acid (DOPAC) in the amygdaloid cortex of rats [129]. Thus, gut bacteria affect central neurotransmission, as 5-HIAA is the major metabolite of serotonin and DOPAC is a metabolite of the neurotransmitter dopamine. Furthermore, gut bacteria were shown to deconjugate host-produced catecholamines via the β-glucuronidase enzyme pathway and thus generate free luminal serotonin and increase concentrations of catecholamines such as noradrenaline and dopamine in mice [130]. These findings reveal that specific gut microbes affect the eating behavior of their hosts through serotonergic signaling or could affect the dopaminergic mesolimbic rewards circuit, which is highly involved in food reward and impulsive choice [37,131]. The oral administration of the probiotic strain *L. acidophilus* NCFM increased intestinal expression of cannabinoid and opioid receptors in mice and rat intestines [132]. In mice, *Lactobacillus rhamnosus* regulated central GABA receptor expression in a vagus nerve-dependent manner [133] and increased central concentrations of neurotransmitter glutamate and its precursor glutamine in addition to N-acetyl aspartate and GABA [134]. The effect on central GABA might be mediated by SCFA since intraperitoneal administration of labeled acetate was shown to cross the BBB, resulting in labeling of the glutamate–glutamine and GABA neuroglial cycles. The signaling cascade went beyond GABA by changing the expression of hypothalamic neurons [135]. A commercially available probiotic consisting of lactobacilli and bifidobacteria from eight different strains decreased the feed intake of mice, accompanied by increased blood butyrate and GLP-1 levels. The finding that, in addition, the gene expression of the hunger-inducing AgRP and NpY was significantly reduced, while the expression of the satiety gene POMC was strongly upregulated, is further evidence that the intestinal microbiota can modulate central mechanisms of feed intake in the hypothalamus [135,136]. Acute acetate administration reduced hypothalamic AMPK activity, thereby leading to increased activity of acetyl-CoA carboxylase. This was shown to elevate malonyl-CoA, which could stimulate the expression of POMC and CART and decrease NPY and AgRP, leading to a reduction in feed intake [135]. As AMPK is the antagonist of mTOR, the question arises of whether mTOR can also be affected by peripheral SCFA availability. Since oral feeding of SCFAs activates mTOR in intestinal cells of mice, and intraperitoneal SCFA administration affects the central regulation [137], it is likely that changes in peripheral SCFA availability could activate central mTOR, with also implications for long-term control of appetite.

Exposure of murine enteroendocrine cells or intestinal organoids to physiological concentrations of SCFAs increased mRNA levels of the umami taste receptors TASR1 and TASR3 [51]. Additionally, there is further evidence from rodent models that the gut microbiota can influence host taste perception and feed selection [30,138]. For example, rats that were prone to increased saccharin consumption differed markedly in the composition of their gut microbiota from rats that were less prone to saccharin consumption [139]. Germ-free mice had altered taste receptors for fat on their tongues and in their intestines [140] or showed increased sucrose intake and had greater numbers of sweet taste receptors in the gastrointestinal tract compared to mice harboring a conventional microbiota [141]. Changes in taste receptor expression and activity have been reported to alter satiety and food preferences [30]. As feedback to taste sensing, taste receptor cells express the anorexigenic hormones GLP-1, PYY and CCK, indicating a peripheral signaling pathway via gastrointestinal hormone secretion, resulting in decreased appetite [142,143]. Accordingly, PYY knockout mice have a decreased behavioral response to both fat- and bitter-tasting compounds [144]. Further, serotonin release is also evoked by bitter, sweet and umami taste stimuli [145,146]. Serotonin is considered the transmitter of most taste cells making synapses with the gustatory nerve. Therefore, the overall view of the evidence available today shows that serotonin appears to be the central neurotransmitter in the regulation of food intake and, together with SCFA, represents the main manipulation mechanism of the GIT microbiota on their host.

## 5. Discussion

According to the holobiont theory of Berg et al. [1], host and microbiota influence each other in a reciprocal manner, which can ultimately lead to behavioral changes in the host. First, the composition of bacteria in the gut depends on the host food base and environmental factors such as stress and animal community. Second, bacteria in the gut seem to have established mechanisms for influencing their hosts, of which the most described are the SCFA; SCFA-mediated secretion of incretins; the metabolism of tryptophan; and the neurotransmitters serotonin, dopamine and GABA. However, the scientific community lacks knowledge on how and to which extent harmful metabolites of microbial fermentation, such as ammonia and hydrogen sulfite, affect the host. Third, eating should imply both homeostasis (sensation of satiety) and hedonics (pleasure). Both experiences are associated with processes that can be influenced by bacteria. The compiled evidence clearly implies that the gut microbiota is an important determinant of host appetite and metabolism. However, the majority of research has been focused on rodent models in the context of human metabolic disorders. As such, less is known about the role of gut microbiota on the modulation of appetite in farm animals. However, in poultry, pig, horse and ruminants, some impact of the GIT microbiota on emotional, social and eating behavior is also described [147]. Continuing research in order to gain a better understanding of the mechanisms in how the GIT microbiota contributes to the modulation of appetite and satiety could have significant consequences for practical animal nutrition. Presently, animal nutrition is based on the estimation of nutrient requirements and the ability of feeds to cover these needs. If we assume in the future that the gut microbiota also modulates appetite and satiety, as has been shown in rodents, this could have major implications for accounting for food preferences and intake in farm animals. However, germ-free rodents are poorly suited as a model for farm animals, and it is a challenge to translate the results to a diverse farm environment. A feasible approach would be to first examine whether the findings obtained in rodents can be transferred to farm animals without (or with what) limitations in order to define further concrete research needs in farm animals. As a future perspective, dietary guidelines for farm animals could be concretized by considerations of the bacterial impact on digestive signaling related to satiety and taste and the peptides produced by bacteria that may be involved in the hypothalamic regulation of appetite. Finally, uncovering the mechanisms by which the microbiome and host interact with each other in terms of appetite regulation could help manage feed intake, homeostasis and gluttony, as well as feed aversion and anorexia related to disease states in farm animals.

## 6. Conclusions

For animal production, feeding behavior and feed intake are essential. Therefore, deeper knowledge is required on how to manipulate GIT microbiota to condition feed intake regulation. Future research should include a focus on how and at which age or (patho-) physiological condition a microbiota manipulation could be feasible and promising in the long term for the animal. Even if we know that the pen community has more impact than heritability or maternal transfer, the impact of early microbial establishment on a further period in animal life needs further elucidation. Currently, the research in microbial control of farm animal behavior is too fragmentary in order to draw concrete strategies for the improvement of feed intake by manipulation of gut microbiota. However, for certain cases, theoretical conclusions can be given. For instance, among farm animals, weaned piglets are probably the most sensitive groups in terms of feed intake, depression and consequences such as diarrhea. Targeted enrichment of the neonatal colon with such microbes associated with high feed intake could represent such a strategy to facilitate the weaning period. Whether this could then be implemented via maternal feeding with probiotics or prebiotics or the supplementary feeding of the suckling piglets must be scientifically verified regarding a sustainable modulation of the individual or at least the groupwise microbial profile. Other starting points include the acidosis treatment in cattle described under 4.2. Instead of rumen liquor transplantation, a targeted administration of beneficial bacteria could also have a preventive effect in early states of acidosis. It would also be conceivable to develop special antibiotic-associated feeds that counteract collateral elimination of beneficial bacteria, similar to what is used in humans today. Ultimately, there is also the question of whether farm animals should be fed in such a way that they have a health-promoting microbiota or whether the microbiota should be manipulated so that the animals eat more or utilize their feed better and grow faster. The answer to this question will probably differ in the different regions of the world with regard to the currently most important social challenges—be it the fight against hunger in some parts of the world or the protection of the environment and animal welfare in regions where there is nutrient oversaturation.

## Figures and Tables

**Figure 1 microorganisms-10-01305-f001:**
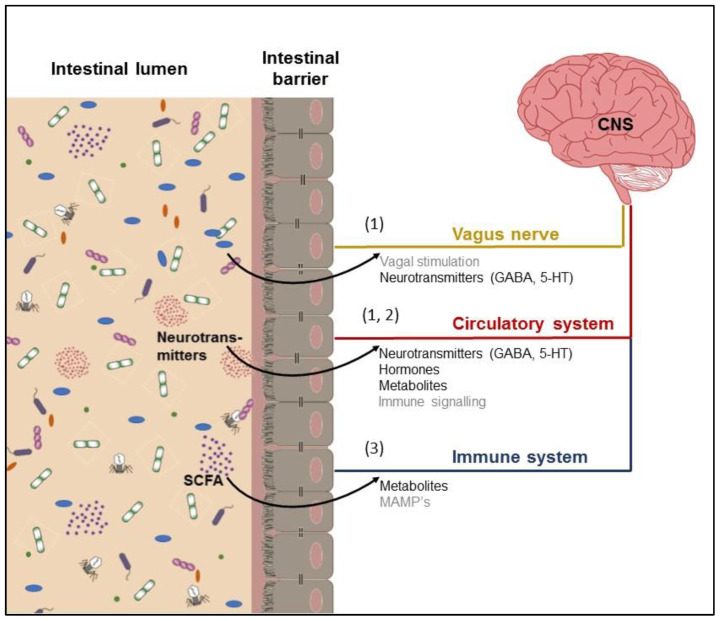
Pathways linking the microbiome and the central nervous system (CNS).

**Table 1 microorganisms-10-01305-t001:** Taxonomic profiles of major gut bacterial communities at the phylum level in farm animals.

Host	Gut Segment	Phylum				References
		Firmicutes	Bacteroidetes	Actinobacteria	Proteobacteria	
Cattle	Rumen	25–58%	38–75%	<1%	0–5%	[14]
Sheep	Rumen	49%	47%	<1%	<1%	[15]
Horse	Cecum	30–50%	30–50%	-	5%	[16]
Rabbit	Cecum	83%	6%	<1%	<1%	[17]
Pig	Colon	54%	42%	<1%	2%	[18]
Chicken	Ceca	44–50%	23–46%	6%	1–16%	[19]
Duck	Ceca	34%	57%	-	7%	[13]

## Data Availability

Not applicable.

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
