# Peer review of "Influence of the Gut Microbiome on Feed Intake of Farm Animals"

_microorganisms, 2022, doi:10.3390/microorganisms10071305_

Round 1
Reviewer 1 Report
Review of Manuscript ID: microorganisms-1768079„Influence of the gut microbiome on feed intake of farm animals”
The manuscript the collects the current knowledge on the connections between the gut microbiome and the feed intake of farm animals. Recognition of the mechanisms by which the microbiome and host interact with each other in terms of appetite regulation could help managing feed intake. Among farm animals, weaned piglets are probably one of the most sensitive groups in terms of too low feed intake and too high tendency to diarrhea. Do the Authors already see any possibility of using the connections described in the manuscript to support the health status of weaned piglets?
The work is well written, divided into logical chapters that organize the content. Only minor suggestions before publication:
L97-99: delete these sentence.
L100: „3. The control of food/feed intake” – delete the „food” as the article is about the farm animals.
Figure 1: add caption/title next to 1 upper arrow (from intestinal lumen to vagus nerve). Add numbers 1,2 and 3 near the adequate paths on the graph, the same as described under the graph.

Author Response
Dear Reviewer 1,
thanks a lot for the kind support and your valuable comments. Please find below the point-to-point response!
The manuscript the collects the current knowledge on the connections between the gut microbiome and the feed intake of farm animals. Recognition of the mechanisms by which the microbiome and host interact with each other in terms of appetite regulation could help managing feed intake. Among farm animals, weaned piglets are probably one of the most sensitive groups in terms of too low feed intake and too high tendency to diarrhea. Do the Authors already see any possibility of using the connections described in the manuscript to support the health status of weaned piglets? --> Thanks for this useful suggestion which is now adressed in L582-588.
The work is well written, divided into logical chapters that organize the content. Only minor suggestions before publication:
L97-99: delete these sentence. DONE.
L100: „3. The control of food/feed intake” – delete the „food” as the article is about the farm animals. DONE.
Figure 1: add caption/title next to 1 upper arrow (from intestinal lumen to vagus nerve). Add numbers 1,2 and 3 near the adequate paths on the graph, the same as described under the graph. DONE.
Kind regards and all the best.
Reviewer 2 Report
This article systematically summarizes the effects of gut microbes on short- and long-term host appetite behavior. As a symbiotic group with the host, the gut microbiota plays an important role in the body's metabolism and intestinal immunity. Studying the relationship between the gut microbiota and the gut is a hotspot of current research. Overall, the manuscript is well written and the logical framework is well organized. It is recommended that this manuscript be published in microorganisms with minor revisions.
Specific comments/suggestions are given below:
1. Line 4:Please change 14195 Berlin to Berlin 14195.
2. Line 6:Please mark Correspondence.
3. Line 57:Please modify .[8]. to [8].
4. Line 97:Please bold “Table 1”.
5. Line 194:Please bold “Figure 1”.
Author Response
Dear Reviewer 2,
thanks a lot for the kind support and your valuable comments. Please find below the point-to-point response!
This article systematically summarizes the effects of gut microbes on short- and long-term host appetite behavior. As a symbiotic group with the host, the gut microbiota plays an important role in the body's metabolism and intestinal immunity. Studying the relationship between the gut microbiota and the gut is a hotspot of current research. Overall, the manuscript is well written and the logical framework is well organized. It is recommended that this manuscript be published in microorganisms with minor revisions.
Specific comments/suggestions are given below:
- Line 4:Please change 14195 Berlin to Berlin 14195. DONE.
- Line 6:Please mark Correspondence. The Hyperlink was removed. Besides that, the correspondence statement should be according to the journals guidelines.
- Line 57:Please modify .[8]. to [8]. Thanks a lot! It is corrected now.
- Line 97:Please bold “Table 1”. Table 1 is set to be bold.
- Line 194:Please bold “Figure 1”. Figure 1 is set to be bold.
Kind regards an all the best!
Reviewer 3 Report
The present review deals with a very interesting topic and with a high potential impact either for applied nutritionists as well as for the academia. The main important items are appetite regulation and feeding behavior driven by microbiota in most of the animal species and humans. Therefore, since for animal production the feeding behavior and feed intake is essential it would be very interesting a deep discussion section integrating a section on how it could be changed GIT microbiota to condition feed intake regulation. This should include how, when (age dependent), the impact of early microbial establishment on further period in animal life?
1. Is that maternal microbial population determinant on the impact of feed regulation of the offspring. In swine, for example the different feeding behavior between half-siblings might be explained by significant microbial difference.
2. Conclusion section is too long and not specific. This is not a conclusion section. Please rename the section as discussion section with the aim of integrating the previous above-mentioned items affecting the regulation.
3. Moreover, since this is strongly important for animal production, please include a section that could be a closing paragraph of the discussion section providing the implication and providing details and review on how microbiota could be changes via diet, feed or other techniques aiming to improve feed intake regulation.
4. Improve the general discussion and rate the impact of the GIT microbiota on animal requirements. At least for the essential nutrients determining growth and efficiency. State if there is any interaction between feeding behavior, GIT microbiota and nutrient requirements. Since feed formulations is a compromise between requirements and feed intake so what are the dynamic implications for feed formulation?
Author Response
Dear Reviewer 3,
thanks a lot for your valuable comments! Please find below the point-by-point response. I hope that I have been able to answer them satisfactorily, even if it was not possible to be completely concrete since too many facts are still unknown.
The present review deals with a very interesting topic and with a high potential impact either for applied nutritionists as well as for the academia. The main important items are appetite regulation and feeding behavior driven by microbiota in most of the animal species and humans. Therefore, since for animal production the feeding behavior and feed intake is essential it would be very interesting a deep discussion section integrating a section on how it could be changed GIT microbiota to condition feed intake regulation. This should include how, when (age dependent), the impact of early microbial establishment on further period in animal life? Thanks a lot for this important suggestion! According to one of your following comments, the previous „Conclusions” section was renamed to „Discussion” and a new „Conclusion” section was created. The new Conclusion contains potential strategies to change GIT microbiota to condition feed intake regulation, as much as can be derived from the rare knowledge on the field existing today.
- Is that maternal microbial population determinant on the impact of feed regulation of the offspring. In swine, for example the different feeding behavior between half-siblings might be explained by significant microbial difference. Addressed in L577-579
- Conclusion section is too long and not specific. This is not a conclusion section. Please rename the section as discussion section with the aim of integrating the previous above-mentioned items affecting the regulation. Thanks for the suggestion for improvement of the manuscript! The section is renamed to Discussion. However, I hope for your patience that I integrated the previous above-mentioned items into the new conclusion section, which is now clearer in show potential practical implications.
- Moreover, since this is strongly important for animal production, please include a section that could be a closing paragraph of the discussion section providing the implication and providing details and review on how microbiota could be changes via diet, feed or other techniques aiming to improve feed intake regulation. Addressed in the new Conclusion section.
- Improve the general discussion and rate the impact of the GIT microbiota on animal requirements. At least for the essential nutrients determining growth and efficiency. State if there is any interaction between feeding behavior, GIT microbiota and nutrient requirements. Since feed formulations is a compromise between requirements and feed intake so what are the dynamic implications for feed formulation? The only species and nutrients we know in terms of requirement are the pigs and amino acids (Torrallardona et al., 2003) as described in the according section 4.1.2. I totally agree that we need to know in which extent bacteria (and certain bacterial patterns) contribute the coverage of nutrient requirements, since bacteria produce also notable amounts of vitamins. However, the available data is rare. Therefore, a main objective in preparing the review manuscript was to summarize the existing knowledge and identify the gaps to drive targeted research in underrepresented farm animals. I hope for your understanding that I cannot address this proposal better at this time and hope you agree with the changes made.
Kind regards and all the best!